# A Predictive Factor Analysis of Social Biases and Task-Performance in Pretrained Masked Language Models

**Yi Zhou**[♠]        **Jose Camacho-Collados**[♠]        **Danushka Bollegala**[†,♣]

Cardiff University[♠], University of Liverpool[♣], Amazon[†]

{Zhouy131,CamachoColladosJ}@cardiff.ac.uk
danushka@liverpool.ac.uk

## Abstract

Various types of social biases have been reported with pretrained Masked Language Models (MLMs) in prior work. However, multiple underlying factors are associated with an MLM such as its model size, size of the training data, training objectives, the domain from which pretraining data is sampled, tokenization, and languages present in the pretrained corpora, to name a few. It remains unclear as to which of those factors influence social biases that are learned by MLMs. To study the relationship between model factors and the social biases learned by an MLM, as well as the downstream task performance of the model, we conduct a comprehensive study over 39 pretrained MLMs covering different model sizes, training objectives, tokenization methods, training data domains and languages. Our results shed light on important factors often neglected in prior literature, such as tokenization or model objectives.

## 1   Introduction

Masked Language Models (MLMs) have achieved promising performance in many NLP tasks (Devlin et al., 2019; Liu et al., 2019; Liang et al., 2023). However, MLMs trained on massive amounts of textual training data have also been found to encode concerning levels of social biases such as gender and racial biases (Kaneko and Bollegala, 2019; May et al., 2019; Dev et al., 2020; Silva et al., 2021; Kaneko et al., 2022). In spite of the overall success of MLMs across NLP tasks, such biases within MLMs raise ethical considerations and underscore the need for debiasing methods to ensure fair and unbiased MLMs.

On the other hand, MLMs are trained by considering and optimising various underlying factors that contribute to their performance on downstream tasks. These factors include but are not limited to parameter size, tokenization methods, training objectives and training corpora. The performance of MLMs is affected by the interplay of such factors.

Nevertheless, it remains unclear as to how these factors influence social biases in MLMs and their downstream task performance.

Evaluating the impact of these factors is challenging due to three main reasons: (a) The factors that we consider within a model are not independent, rather, they exhibit complicated interdependence and affect the performance of models simultaneously. (b) MLMs are diverse with different architectures, configurations and parameters. The diversity across models requires the need for generalisation and abstraction when considering the values of factors. (c) Many recent works proposed debiasing methods to mitigate social biases in MLMs (Webster et al., 2020; Lauscher et al., 2021; Schick et al., 2021; Guo et al., 2022). However, most debiasing methods tend to worsen the performance of MLMs in downstream tasks (Meade et al., 2022). Therefore, it is crucial to consider the trade-off between social bias and downstream task performance when comparing MLMs.

To address the non-independent issue of factors, we propose a method using Gradient Boosting (Freund and Schapire, 1997) to consider dependencies among factors. Moreover, we use the coefficient of determination ($R^2$; Nagelkerke et al., 1991) as a measure to analyse the importance of factors. Regarding the diversity of MLMs, we converge a broad set of MLMs across multiple languages and 4 domains, resulting in 39 MLMs for evaluation in total. Moreover, we incorporate TweetEval and GLUE to evaluate the downstream task performance of MLMs, meanwhile, evaluating their social biases intrinsically using All Unmasked Likelihood with Attention weights (AULA; Kaneko and Bollegala, 2022) on two benchmarks StereoSet (SS; Nadeem et al., 2021) and crowdsourced stereotype pairs benchmark (CP; Nangia et al., 2020a). Note that we are not proposing novel bias evaluation measures in this paper. Instead, we use existing metrics such as AULA to evaluate social biases.

Our experimental results indicate that model size, training objectives and tokenization are the three most important categories of factors that affect the social bias and downstream task performance of MLMs. Interestingly, we observe that models using Byte-Pair Encoding (BPE; Sennrich et al., 2016) include lower level of social biases, while achieving the best downstream performance compared to models using other tokenization methods. Overall, multilingual models tend to have less biases than their monolingual counterparts.

## 2   Related Work

As MLMs have been successfully applied to diverse NLP tasks, it is important to study the factors that determine their social biases. Rogers et al. (2020) reviewed the current state of knowledge regarding how BERT works, what kind of information it learns and how it is encoded, typically alterations to its training objectives and architecture, the overparameterization issue and approaches to compression. Xia et al. (2020) studied contextualised encoders in various aspects and discussed the trade-off between task performance and the potential harms contained in the pretraining data. Later, Pérez-Mayos et al. (2021) investigated the effect of pretraining data size on the syntactic capabilities of RoBERTa and they showed that models pretrained with more data tend to encode better syntactic information and provide more syntactic generalisation across different syntactic structures. However, these studies focus on MLMs in downstream tasks, while none consider the social biases in MLMs.

On the other hand, models trained for different downstream tasks have been found to exhibit social biases. Kiritchenko and Mohammad (2018) evaluated gender and racial biases across 219 automatic sentiment analysis systems and discovered statistically significant biases occurring in several systems. Díaz et al. (2018) studied age-related biases in sentiment classification and discovered that significant age bias is encoded in the output of many sentiment analysis systems as well as word embeddings.

Zhao et al. (2020) focused on gender bias in multilingual embeddings and its influence on the process of transfer learning for NLP applications. They showed that the level of bias in multilingual representations varies depending on how the embeddings are aligned to different target spaces, and that the alignment direction can also affect the bias

in transfer learning. Choenni et al. (2021) investigated the types of stereotypical information that are captured by pretrained language models and showed the variability of attitudes towards various social groups among models and the rapid changes in emotions and stereotypes that occur during the fine-tuning phase.

Existing bias evaluation methods use different strategies such as pseudo likelihood (Kaneko and Bollegala, 2022), cosine similarity (Caliskan et al., 2017; May et al., 2019), inner-product (Ethayarajh et al., 2019), to name a few. Independently of any downstream tasks, intrinsic bias evaluation measures (Nangia et al., 2020b; Nadeem et al., 2021; Kaneko and Bollegala, 2022) evaluate social biases in MLMs stand alone. However, given that MLMs are used to represent input texts in a variety of downstream tasks, multiple previous works have proposed that social biases should be evaluated with respect to those tasks (De-Arteaga et al., 2019; Webster et al., 2020). Kaneko and Bollegala (2021) showed that there exists only a weak correlation between intrinsic and extrinsic social bias evaluation measures. In this paper, we use an intrinsic bias evaluation measure, namely AULA, to evaluate social biases in MLMs. AULA has been shown to be the most reliable bias evaluation measure (Kaneko and Bollegala, 2022), hence we use it as our bias evaluation measure.

Although we specifically focus on MLMs in this paper, evaluating the performance predictors for Neural Architecture Search (NAS) (White et al., 2021; Elsken et al., 2019) has been conducted for designing high performance neural networks. Generalisability of the identified factors is an important in NAS because the selected neural architecture would be trained on different datasets to obtain different models. Although it would be ideal to perform a controlled training of MLMs where we experimentally fix all other factors except for a single target factor and then analyse the trained MLM, this is an expensive undertaking given the computational cost of training MLMs on large datasets. On the other hand, we consider already pre-trained MLMs that are publicly made available and do not train any MLMs during this work.

## 3   Analysis of Factors

In order to study the impact of different factors in MLMs, we consider 30 factors and split them into 5 categories. The details of each individual factor

are provided in Appendix C.

## 3.1 Model Size

Models with smaller sizes are generally more lightweight and require less computational resources, making them suitable for deployment on resource-constrained devices or in environments with limited processing power. However, larger models tend to have better performance on downstream tasks but demand more memory, computational power, and longer inference times. On the other hand, the different architectures of models have various numbers of layers as well as training parameters. Recently, MLMs have achieved impressive results on downstream tasks by scaling model size or training with larger datasets (Conneau et al., 2020; Goyal et al., 2021; Liang et al., 2023). To investigate the impact of model size, we consider 3 factors: (1) parameter size, (2) number of layers and (3) vocabulary size.

The parameter size is considered as a categorical feature, in which we divide the parameter size of an MLM into 3 categories according to pre-defined ranges. Specifically, we assign S if the parameter size of an MLM is less than 100M, M if the size is within 100M-300M, and L if the size is greater than 300M. Similarly, we convert the vocabulary size into the same three categories for the models with vocabulary sizes less than 50K, within 50K-100K and greater than 100K, respectively. For the number of layers, we use the number as a feature: 6, 12 and 24 layers.

## 3.2 Training Methods and Objectives

In this category, we consider the methods used during model pretraining as well as the training objectives of MLMs. First, we take into account different masking techniques, starting with the masking technique initially proposed for training BERT (Devlin et al., 2019). Masked language modelling is an objective used during pretraining to improve the model's understanding, in which a certain percentage of words in the input text are randomly selected and replaced with a special [MASK] token, then the model is trained to predict the original word based on its context. Later, they further proposed whole word masking, which aims to improve the handling of individual words with a context. Rather than randomly selecting WordPiece (Wu et al., 2016) produced subtokens to mask, whole word masking always masks the entire words at once, which has been shown to reduce ambiguity and enable

better word-level comprehension and contextual understanding for MLMs (Cui et al., 2021). Apart from these two masking techniques, we consider three other training objectives: (a) next sentence prediction, (b) sentence ordering prediction, and (c) mention reference prediction. We consider the training objectives to be binary and assign 1 if each of them is used and 0 otherwise.

Model distillation is a training technique aiming to train a small student model to transfer the knowledge and capabilities from a larger teacher model (Hinton et al., 2015). This technique has been shown to effectively compact a large language model, while retaining comparable performance compared to the original model (Sanh et al., 2019; Wu et al., 2022). Model distillation is regarded as a binary factor, which is assigned 1 if an MLM uses model distillation, otherwise, it returns 0.

## 3.3 Training Corpora

Training corpora are the collections of texts or data used to train MLMs. According to Kalyan et al. (2021), training corpora can be classified into four types: (1) general, (2) social media, (3) language-specific and (4) domain-specific. In order to conduct a comprehensive study of MLMs trained using different types of training corpora, we cover all four types of training corpora, resulting in four different domains (including general domain): (1) General Domain: BookCorpus (Books), Wikipedia (Wiki), Common Crawl-News (CCNews), OpenWebText (OWT), and Stories; (2) Social Media: Tweets and the Reddit Abusive Language English dataset (RALE); (3) Legal Domain: Patent Litigations, Caselaw Access Project (Caselaw) and Google Patents Public Data (GooglePatents); (4) Biomedical Domain: Full-text biomedical papers from the Semantic Scholar Open Research Corpus (S2ORC), PubMed Abstracts (PMed), PMC Full-text articles and the Medical Information Mart for Intensive Care III (MIMIC3); Finally, we also consider a multilingual corpus: CommonCrawl Corpus in 100 languages (CC100). Each of the training corpora is considered as an individual binary factor.

Owing to the domain of an MLM being associated with the training corpora sampled in that certain domain, we additionally consider domain as a separate factor. This domain factor is included as a categorical factor, with 4 different domains as categories: general domain, social media, legal domain and biomedical domain. Finally, we take

into account continuous training as a binary factor which takes 1 if an MLM is continuously trained on training corpora from different domains and 0 if the model is trained from scratch.

### 3.4 Tokenization

Tokenization is an essential process of breaking down a sequence of text into smaller units, which is able to convert unstructured text data into a format that can be processed by MLMs. Prior works study different tokenization methods on MLMs in different languages (Park et al., 2020; Rust et al., 2021; Toraman et al., 2023), however, the impact of tokenization methods on social bias in MLMs and different downstream tasks remains unrevealed. Therefore, we consider the three most commonly used tokenization methods as categorical factors: BPE, WordPiece and SentencePiece.

### 3.5 Language

In this category, we consider both monolingual and multilingual MLMs. Specifically, we regard language as a categorical factor and use *English* and *Multilingual* to represent if an MLM is monolingual or multilingual, respectively. In addition, the number of languages is also considered as a separate factor, which is categorical and takes the actual number of languages an MLM trained on.

## 4 Masked Language Models

To conduct a comprehensive study of different factors of MLMs affecting social bias and task performance, we evaluate 39 pretrained MLMs,[1] which we divide into four categories as follows. Table 1 summarizes all models.

**Monolingual and general domain models**  We take the monolingual MLMs with different settings pretrained in general domain, including RoBERTa (Liu et al., 2019), BERT (Devlin et al., 2019), ALBERT (Lan et al., 2019), DistilBERT (Sanh et al., 2019), CorefRoBERTa and CorefBERT (Ye et al., 2020).

---

[1]In our initial selection of models, albert-xlarge-v2, nielsr/coref-roberta-large, vinai/bertweet-large, xlm-roberta-base and facebook/xlm-v-base attained an unusual poor performance on the Corpus of Linguistic Acceptability (CoLA) (i.e., a subtask of GLUE) and TweetEval. This is likely caused by an implementation issue that would need some modification. Therefore, to avoid potentially false outliers, we omitted these five models when evaluating TweetEval and GLUE.

| Model Type | Models |
|---|---|
| General Domain (Monolingual) | roberta-base, roberta-large, bert-base-cased, bert-large-cased, bert-base-uncased, bert-large-uncased, bert-large-cased-whole-word-masking, albert-base-v2, albert-large-v2, albert-xlarge-v2, albert-xxlarge-v2, distilbert-base-cased, distilbert-base-uncased, distilroberta-base, bert-large-uncased-whole-word-masking, nielsr/coref-roberta-large, nielsr/coref-roberta-base, nielsr/coref-bert-base, nielsr/coref-bert-large |
| Domain-specific (Monolingual) | cardiffnlp/twitter-roberta-base, cardiffnlp/twitter-scratch-roberta-base, vinai/bertweet-base, vinai/bertweet-large, GroNLP/hateBERT, al-lenai/biomed_roberta_base, saibo/legal-roberta-base, emilyalsentzer/Bio_ClinicalBERT, dmis-lab/biobert-base-cased-v1.2, bionlp/bluebert_pubmed_mimic_uncased_L-12_H-768_A-12, bionlp/bluebert_pubmed_mimic_uncased_L-24_H-1024_A-16, cardiffnlp/twitter-roberta-large-2022-154m, cardiffnlp/twitter-roberta-base-2022-154m |
| General Domain (Multilingual) | bert-base-multilingual-cased, bert-base-multilingual-uncased, distilbert-base-multilingual-cased, xlm-roberta-base, xlm-roberta-large, xlm-v-base |
| Domain-specific (Multilingual) | cardiffnlp/twitter-xlm-roberta-base |

Table 1: Summary of the MLMs in the analysis.

**Monolingual and domain-specific models**  We consider the MLMs either directly or continuously trained in domain-specific English corpora with different settings, consisting of RoBERTa in social media domain (Barbieri et al., 2020), BERT in social media domain (Nguyen et al., 2020; Caselli et al., 2021), RoBERTa in legal domain (Geng et al., 2021), RoBERTa in biomedical domain (Gururangan et al., 2020), and BERT in biomedical domain (Alsentzer et al., 2019; Lee et al., 2020; Peng et al., 2019).

**Multilingual and general domain models**  We take into account the multilingual MLMs with different settings pretrained in the general domain, containing different settings of multilingual BERT (Devlin et al., 2019), DistillBERT (Sanh et al., 2019) and XLM-R (Conneau et al., 2020).

**Multilingual and domain-specific models**  We select the multilingual MLMs pretrained in domain-specific corpora, containing XLM-T (Barbieri et al., 2022).

## 5 MLM Evaluation Metrics and Tasks

Our goal in this paper is to study the relationship between model factors and social bias as well as downstream task performance in pretrained MLMs. For this purpose, we conduct our evaluations using three different types of tasks.

## 5.1 Social Bias

For an MLM under evaluation, we compare the pseudo-likelihood scores returned by the model for stereotypical and anti-stereotypical sentences using AULA (Kaneko and Bollegala, 2022). AULA evaluates social biases by taking into account the MLM attention weights as the importance of tokens. This approach is shown to be robust against frequency biases in words and offers more reliable estimates compared to alternative metrics used to evaluate social biases in MLMs.

Following the standard evaluation protocol, we provide AULA the complete sentence $S = t_1, \ldots, t_{|S|}$, which contains a length $|S|$ sequence of tokens $t_i$, to an MLM with pretrained parameters $\theta$. We compute the Pseudo Log-Likelihood, denoted by $\text{PLL}(S)$, to predict all tokens in $S$ excluding begin and end of sentence tokens. The PLL(S) score of sentence $S$ given by (1) can be used to evaluate the preference expressed by an MLM for $S$:

$$\text{PLL}(S) := \frac{1}{|S|} \sum_{i=1}^{|S|} \alpha_i \log P(t_i | S; \theta) \quad (1)$$

Here $\alpha_i$ is the average of all multi-head attention weights associated with $t_i$, while $P(t_i | S; \theta)$ is the probability assigned by the MLM to token $t_i$ conditioned on $S$.

Given a sentence pair, the percentage of stereotypical ($S^{st}$) sentence preferred by the MLM over anti-stereotypical ($S^{at}$) one is considered as the AULA *bias score* of the MLM and is given by (2):

$$\text{AULA} = \left( \frac{100}{N} \sum_{(S^{st}, S^{at})} \mathbb{I}(\text{PLL}(S^{st}) > \text{PLL}(S^{at})) \right) \quad (2)$$

Here, $N$ is the total number of text instances and $\mathbb{I}$ is the indicator function, which returns 1 if its argument is True and 0 otherwise. AULA score given by (2) falls within the range $[0, 100]$ and an unbiased model would return bias scores close to 50, whereas bias scores less or greater than 50 indicate bias directions towards the anti-stereotypical or stereotypical group, respectively.

**Social Bias Benchmarks** We conduct experiments on the two most commonly used social bias evaluation datasets for MLMs: StereoSet (SS) and Crowdsourced Stereotype Pairs benchmark (CP). SS contains associative contexts, which cover four types of social biases: race, gender, religion, and profession, while CP is crowdsourced and annotated by workers in the United States, consisting of nine types of social biases: race, gender, sexual orientation, religion, age, nationality, disability, physical appearance, and socioeconomic status/occupation. We follow the work from Kaneko and Bollegala (2022)[2] and use the default setting for evaluation. We denote the AULA computed on the CP and SS datasets by A-CP and A-SS, respectively, in the rest of the paper.

## 5.2 Downstream Performance

To further investigate the impact of factors of an MLM in terms of its downstream tasks performance, we evaluate MLMs on two additional benchmark datasets: The General Language Understanding Evaluation (GLUE) benchmark (Wang et al., 2018)[3] and social media TweetEval benchmark (Barbieri et al., 2020).[4]

**GLUE** GLUE is comprised of 9 tasks for evaluating natural language understanding systems. The tasks in GLUE are Corpus of Linguistic Acceptability (CoLA; Warstadt et al., 2019), Stanford Sentiment Treebank (SST-2; Socher et al., 2013), Microsoft Research Paraphrase Corpus (MRPC; Dolan and Brockett, 2005), Semantic Textual Similarity Benchmark (STS-B; Cer et al., 2017), Quora Question Pairs (QQP; Iyer et al., 2017), Multi-Genre NLI (MNLI; Williams et al., 2018), Question NLI (QNLI; Rajpurkar et al., 2016), Recognizing Textual Entailment (RTE; Dagan et al., 2005; Haim et al., 2006; Giampiccolo et al., 2007; Bentivogli et al., 2009) and Winograd NLI (WNLI; Levesque et al., 2012). These tasks are framed as classification tasks for either single sentences or pairs of sentences. We follow the finetuning procedures from prior work (Devlin et al., 2019; Liu et al., 2019) and report results on the development sets.

**TweetEval** TweetEval is a unified Twitter benchmark composed of seven heterogeneous tweet classification tasks. The tasks in TweetEval are emoji prediction (Barbieri et al., 2018), emotion recognition (Mohammad et al., 2018), hate speech detection (Basile et al., 2019), irony detection (Van Hee et al., 2018), offensive language identification (Zampieri et al., 2019), sentiment

---

[2] https://github.com/kanekomasahiro/evaluate_bias_in_mlm

[3] https://gluebenchmark.com/

[4] https://github.com/cardiffnlp/tweeteval

analysis (Rosenthal et al., 2017) and stance detection (Mohammad et al., 2016). We use the default setting as the TweetEval original baselines to fine-tune pretrained MLMs on the corresponding training set and report average results on the test set after 3 runs.

## 5.3 Correlation between tasks

Table 2 shows the Pearson and Spearman's correlation coefficients between each pair of task performances over the 39 MLMs. We observe a moderate correlation between A-SS and GLUE, which indicates that better performance in GLUE entails a higher stereotypical bias in SS (this is also true for CP but to a much lesser extent). In contrast, there is no significant correlation observed between the models' performance on downstream tasks, i.e., TweetEval vs. GLUE.

| Task pair | Pearson | Spearman |
|---|---|---|
| A-CP vs. A-SS | $0.607^\dagger$ | $0.623^\dagger$ |
| A-CP Vs. TweetEval | 0.193 | 0.214 |
| A-CP vs. GLUE | 0.244 | 0.286 |
| A-SS vs. TweetEval | 0.338 | 0.304 |
| A-SS vs. GLUE | $0.382^\dagger$ | $0.487^\dagger$ |
| TweetEval vs. GLUE | 0.229 | 0.309 |

Table 2: Pearson and Spearman correlations between the models' performance on pairs of tasks, where † denotes statistically significant ($p < 0.05$) correlations.

## 6 Regression Analysis

To investigate the importance of the different factors on social bias and the task performance of MLMs, we train a regression model.

## 6.1 Experimental setting

We generate the features for each MLM according to its factors as described in §3. An example of the features of an individual model is given in Table 3. These features are fed into a regression model as input for training, using both social bias and task performance as output.

In order to select the best regression model for this purpose, we compare the performance of 6 different regression models on each prediction task, namely gradient boosting, support vector machine, gaussian process regression, decision tree, random forest and linear regression. We compute the performance of the regression models by means of their coefficient of determination ($R^2$) and root mean squared error (RMSE) for each prediction task. We use the regression models implemented in sklearn with the default parameters and take the averages over three independent runs.

**Regression model comparison** The comparison of different regression models trained using the features from the monolingual MLMs in the general domain is shown in Table 4, while the performance of regression models trained using the features from all of the MLMs is shown in §A.1. From Table 4, we observe that gradient boosting obtains the best performance in terms of both $R^2$ and RMSE on both A-CP and TweetEval, while the decision tree obtains the best performance on A-SS. Almost all of the regression models return negative $R^2$ scores for GLUE, which proves hard to predict from the analysed factors. An error analysis of each GLUE subtask on gradient boosting in terms of $R^2$ scores is shown in Appendix B. In contrast, the three remaining social-related evaluations, including TweetEval, can be predicted to a reasonable extent. The linear regression model obtains the lowest $R^2$ scores for both A-CP and GLUE, which indicates that a linear model may not be suitable for predicting the performance of MLMs. Given the results, we decided to use gradient boosting for the rest of the experiments in this paper.

**Feature importance** In addition, to investigate the influence of the factors on the performance of MLMs using $R^2$ and RMSE, we compute the importance score of each factor after training the regression model using the Gini importance implemented in sklearn. The Gini importance is computed as the normalized total reduction of the criterion brought by that feature. It calculates the global contribution of each feature by aggregating the losses incurred in each split made by that feature (Delgado-Panadero et al., 2022).

## 6.2 Results

**Feature importance** Table 5 shows the Gini importance scores of each factor. Due to space constraints in this table, we have omitted the factors that received a score of 0 importance (the full table is shown in §A.2). The parameter size obtains the largest importance score for A-CP, while it is the second most important factor for TweetEval and GLUE. Tokenization is the most important factor for A-SS and the second most for A-CP.

| Model | Features |
|---|---|
| bert-large-cased | Parameter Size: L, Distill: 0, Train-Books: 1, Train-Wiki: 1, Train-CCnews: 0, Train-OWT: 0, Train-Stories: 0, Train-Tweets: 0, Train-RALE: 0, Train-Patent: 0, Train-Caselaw: 0, Train-GooglePatents: 0, Train-S2ORC: 0, Train-MIMIC3: 0, Train-PMed: 0, Train-PMC: 0, Train-CC100: 0, Cont-train: 0, Uncased: 0, Domain: general, Language: en, Number of Languages: 1, Vocabulary size: S, Tokenization: WordPiece, Number of Layers: 24, Masked Language Modelling: 1, Whole Word Masking: 0, Next Sentence Prediction: 1, Sentence Order Prediction: 0, Mention Reference Prediction: 0 |

Table 3: Example of features of MLMs given to a regression model for training according to considered factors.

| | Models | A-CP | A-SS | TweetEval | GLUE |
|---|---|---|---|---|---|
| $R^2$ | Gradient Boosting | **0.594** | 0.652 | **0.481** | -0.201 |
| | SVM | 0.033 | 0.164 | 0.135 | -0.053 |
| | Gaussian Process | 0.172 | 0.129 | 0.323 | -0.077 |
| | Decision Tree | 0.229 | **0.776** | 0.072 | -1.399 |
| | Random Forest | 0.434 | 0.554 | -0.193 | -0.093 |
| | Linear Regression | 0.016 | 0.741 | 0.353 | -4.846 |
| RMSE | Gradient Boosting | **2.661** | 2.385 | **1.459** | 1.883 |
| | SVM | 4.107 | 3.696 | 1.886 | **1.764** |
| | Gaussian Process | 3.801 | 3.772 | 1.668 | 1.784 |
| | Decision Tree | 3.667 | **1.913** | 1.847 | 2.663 |
| | Random Forest | 3.143 | 2.701 | 2.214 | 1.798 |
| | Linear Regression | 4.144 | 2.056 | 1.630 | 4.157 |

Table 4: Comparison of different regression models using the features from the monolingual MLMs in the general domain. The highest $R^2$ (on the top) and the lowest RMSE (on the bottom) are shown in bold.

| Factors | A-CP | A-SS | TweetEval | GLUE |
|---|---|---|---|---|
| Parameter Size | **0.297** | 0.134 | 0.244 | 0.200 |
| Distill | 0.007 | 0.001 | 0.005 | 0.002 |
| Train-CCnews | 0.005 | 0.001 | 0.014 | 0.001 |
| Train-OWT | 0.004 | 0.002 | 0.015 | 0.000 |
| Train-Stories | 0.004 | 0.001 | 0.026 | 0.003 |
| Cont-train | 0.037 | 0.001 | 0.072 | 0.007 |
| Uncased | 0.071 | 0.080 | 0.007 | 0.035 |
| Vocabulary Size | 0.005 | 0.000 | 0.016 | 0.001 |
| Tokenization | 0.191 | **0.356** | 0.019 | 0.059 |
| # Layers | 0.056 | 0.047 | 0.093 | **0.520** |
| MLM | 0.136 | 0.115 | 0.024 | 0.008 |
| WWM | 0.162 | 0.169 | 0.015 | 0.013 |
| NSP | 0.001 | 0.041 | 0.125 | 0.047 |
| SOP | 0.002 | 0.052 | **0.256** | 0.100 |
| MRP | 0.023 | 0.001 | 0.071 | 0.005 |

Table 5: The important scores for each of the factors for training the gradient boosting regression model. The highest importance scores are shown in bold, and the second-highest ones are underlined. cont-train, MLM, WWM, NSP, SOP and MRP represent continual training, masked language modeling, whole word masking, next sentence prediction, sentence ordering prediction and mention reference prediction, respectively.

**Factor analysis** To further study the effects of factors on MLMs, we eliminate each of the im-

| Factors | A-CP | A-SS | TweetEval | GLUE |
|---|---|---|---|---|
| *Without Removing* | 0.594 | 0.652 | 0.481 | -0.201 |
| Parameter Size | **0.381** | 0.567 | 0.121 | -0.288 |
| Distill | 0.608 | 0.682 | 0.468 | -0.254 |
| Train-CCNews | 0.594 | 0.641 | 0.488 | -0.222 |
| Train-OWT | 0.594 | 0.639 | 0.464 | -0.218 |
| Train-Stories | 0.598 | 0.640 | 0.499 | -0.228 |
| Cont-train | 0.579 | 0.634 | 0.401 | -0.174 |
| Uncased | 0.590 | **0.278** | 0.373 | -0.420 |
| Vocabulary Size | 0.579 | 0.640 | 0.465 | -0.179 |
| Tokenization | 0.544 | 0.480 | 0.496 | **-0.750** |
| # Layers | 0.508 | 0.596 | 0.440 | 0.056 |
| MLM | 0.601 | 0.645 | 0.483 | -0.239 |
| WWM | 0.604 | 0.642 | 0.382 | -0.134 |
| NSP | 0.594 | 0.765 | 0.732 | -0.612 |
| SOP | 0.604 | 0.567 | **0.083** | -0.641 |
| MRP | 0.602 | 0.639 | 0.349 | -0.146 |

Table 6: $R^2$ scores after removing individual factors. The most important factors on each predicted task (i.e, those factors that lead to a larger $R^2$ drop) are shown in bold, and the second-best ones are underlined. MLM, WWM, NSP, SOP and MRP represent masked language modeling, whole word masking, next sentence prediction, sentence ordering prediction and mention reference prediction, respectively.

portant factors (i.e., the ones that obtain non-zero importance scores) at a time and track the $R^2$ score returned by the gradient boosting models trained on different tasks. Table 6 shows the result. In this table, the lower $R^2$ score indicates the more important the factor is. Consistent with the result shown in Table 5, sentence order prediction and parameter size are the most and second most important factors for TweetEval, respectively, and parameter size is the most important factor for A-CP. For A-SS, uncased and tokenization are the most and second most important factors, respectively. In addition, we show the corresponding important scores for each factor for training decision tree models in §A.3 and observe largely similar conclusions to the result presented in Table 5.

**Types of features** To further explore the impact of a group of factors, we consider the 5 categories

in §3. To investigate the effect of a certain group, we conduct ablations by removing a group of factors as well as retaining only one group at a time. Tables 7 and 8 show the corresponding results. The training objectives group of factors is regarded as paramount for social bias evaluation (i.e., A-CP and A-SS) in Table 7, and their removal leads to a substantial decrease in the $R^2$ scores.

Meanwhile, Table 8 shows much lower $R^2$ scores across all the cases. This is because of the discrepancy between removing a group of factors and retaining only one category, as the latter entails a reduced utilization of features during the training process. Despite this limitation, we observe that by keeping tokenization only, the regression model can still make relatively accurate predictions for social bias evaluation, indicating that tokenization is important for social bias evaluation. In contrast, training corpora and training objectives are the most important ones on the downstream tasks.

| Categories | A-CP | A-SS | TweetEval | GLUE |
|---|---|---|---|---|
| *Without Removing* | 0.594 | 0.652 | 0.481 | -0.201 |
| Model Size | 0.587 | 0.670 | **0.304** | -0.290 |
| Training Corpora | 0.631 | 0.607 | 0.322 | -0.217 |
| Training Objectives | **-1.010** | **-1.300** | 0.415 | -0.222 |
| Tokenization | 0.563 | 0.100 | 0.428 | **-1.210** |
| Language | 0.589 | 0.646 | 0.450 | -0.173 |

Table 7: $R^2$ scores removing features from a single category. The most important categories on each task (i.e., those causing a larger $R^2$ drop) are shown in bold.

| Categories | A-CP | A-SS | TweetEval | GLUE |
|---|---|---|---|---|
| *ALL* | 0.594 | 0.652 | 0.481 | -0.201 |
| Model Size | 0.096 | -0.420 | 0.191 | -0.780 |
| Training Corpora | -0.044 | 0.028 | **0.340** | -0.225 |
| Training Objectives | -0.180 | 0.319 | -0.944 | **0.008** |
| Tokenization | **0.240** | **0.358** | 0.251 | -0.024 |
| Language | 0.000 | -0.004 | -0.488 | -0.005 |

Table 8: $R^2$ scores keeping features from one category only. The most important factors on each predicted task are shown in bold.

## 7 Qualitative Analysis

With the knowledge that model size, tokenization, and training objectives are the three primary categories of factors influencing social bias and task performance in MLMs from §6.2, we further investigate the factors within each category to discern their individual contributions in MLMs. For this purpose, we calculate the average performance of

MLMs when considering a specific feature associated with a factor for each task. To capture the overall trend, we extend our analysis to include not only monolingual MLMs in the general domain but also domain-specific and multilingual MLMs.

| | A-CP | A-SS | TweetEval | GLUE |
|---|---|---|---|---|
| Parameter Size | | | | |
| S ($x < 100M$) | **54.866** | **57.730** | 59.903 | 78.827 |
| M ($100M \le x \le 300M$) | 57.388 | 60.246 | **64.472** | 80.137 |
| L ($x > 300M$) | 59.072 | 59.838 | 64.040 | **81.449** |
| Number of Layers | | | | |
| 6 layers | **53.383** | **55.097** | 60.116 | 76.288 |
| 12 layers | 57.546 | 60.398 | 64.472 | 81.225 |
| 24 layers | 59.336 | 60.104 | **64.714** | **82.163** |
| Vocabulary Size | | | | |
| S ($x < 50K$) | 58.196 | 60.674 | 62.101 | **82.413** |
| M ($50K \le x \le 100K$) | 58.422 | 60.322 | **65.676** | 81.472 |
| L ($x > 100K$) | **52.386** | **53.602** | 59.714 | 74.832 |
| Tokenization | | | | |
| BPE | 58.422 | 60.322 | **65.676** | **81.472** |
| WordPiece | 57.838 | 60.086 | 62.101 | 81.287 |
| SentencePiece | **54.800** | **56.382** | 60.470 | 77.210 |
| Training Objectives | | | | |
| MLM | 57.904 | 60.294 | **65.676** | 81.028 |
| MLM + NSP | 56.142 | 58.034 | 61.593 | 80.197 |
| MLM + SOP | 56.995 | 59.185 | 57.314 | **84.926** |
| MLM + MRP | **52.140** | **56.230** | 60.012 | 79.872 |
| WWM + NSP | 59.715 | 60.805 | 62.225 | 80.420 |

Table 9: Comparison of social bias and task performance of the top 5 performing MLMs on different tasks, according to different features associated with the important factors. MLM, NSP, WWM, SOP and MRP represent masked language modeling, next sentence prediction, whole word masking, sentence ordering prediction and mention reference prediction, respectively.

Table 9 shows the average scores for the top-5 performing MLMs for each category of the important factors. Note that the number of models associated with each category within a specific factor is different. For example, there are 21 MLMs in the *medium* and 12 in the *large* categories, whereas there are only 6 in the *small* category for the parameter size factor. Because some outlier cases are affecting the overall averages, in Table 9 we compute the top-5 performing models in each category, whereas the averages over all models are shown in §A.4.

For the model size, we observe that the MLMs trained with a parameter size greater than 100M tend to have better downstream task performance, while reporting higher levels of social biases compared to the ones with a smaller parameter size. The models trained with 6 layers demonstrate the least amount of social biases (i.e., A-CP and A-SS close to 50), however, the ones trained with 24 lay-

ers obtain better downstream task performance. As for the vocabulary size, the models trained with small- and medium-sized vocabulary obtain the best performance on GLUE and TweetEval, respectively, whereas the ones with a large vocabulary size contain less degree of social biases. Regarding the tokenization methods, we see that the models using SentencePiece contain the lowest level of social biases, while the models using BPE obtain better downstream task performance. With respect to training objectives, the models trained with both masked language modelling and mention reference prediction contain the fewest degrees of social biases. On the other hand, models trained with masked language modelling only and models with both masked language modelling and sentence ordering prediction return the best downstream task performance on TweetEval and GLUE, respectively.

## 8 Discussion

**Model size.** Larger MLMs have a higher capability to achieve better performance on downstream tasks, while smaller models are computationally efficient and require fewer hardware resources. From Table 9, we see that models with a larger number of parameters and more layers report better performance in GLUE and TweetEval compared to the smaller and shallower models, while at the same time demonstrating lesser social biases in terms of A-CP and A-SS scores. Because the learning capacity of MLMs increases with the number of parameters and the depth (i.e. number of layers), it is not surprising that larger models outperform the smaller ones on downstream tasks. However, it is interesting to observe that social biases do not necessarily increase with this extra capacity of the MLMs. In the case of gender-related biases, Tal et al. (2022) showed that even if the gender bias scores measured on Winogender (Rudinger et al., 2018) are smaller for the larger MLMs, they make more stereotypical errors with respect to gender. However, whether this observation generalises to all types of social biases remains an open question.

**Mono- vs. Multi-lingual.** Table 10 top compares the performance of monolingual vs. multilingual MLMs. We see that multilingual models demonstrate lower levels of social biases compared to their monolingual counterparts. This aligns with prior works that conjecture a multilingual language model to benefit from training over a larger number

| Models | A-CP | A-SS | TweetEval | GLUE |
|---|---|---|---|---|
| Monolingual | 54.157 | 56.062 | **60.953** | **79.648** |
| Multilingual | **50.342** | **52.092** | 58.187 | 71.723 |
| General Domain | 54.157 | **56.062** | 60.953 | **79.648** |
| Domain-specific | 54.042 | 56.962 | 61.062 | 75.547 |
| Social Media Domain | **53.325** | 57.438 | **64.339** | 75.713 |

Table 10: The performance of monolingual vs. multilingual models and general domain vs. domain-specific models on social bias and downstream tasks.

of languages, thus incorporating a greater spectrum of cultural diversity (Liang et al., 2020; Ahn and Oh, 2021). Consequently, the presence of these divergent viewpoints within the model can potentially mitigate social biases. Conversely, monolingual models obtain better performance on TweetEval and GLUE, which are limited to English.

**General vs. Domain-specific.** Table 10 bottom shows the performance of models from different domains. Recall from §4 that we include domain-specific models for social media, legal and biomedical domains. As TweetEval is a social media benchmark, we additionally include the performance of models in the social media domain. Models in the social media domain contain the least bias according to A-CP and achieve the best performance on TweetEval. Conversely, the performance of models in the general domain is better than domain-specific ones on GLUE and obtains the lowest bias score for A-SS. This result is not surprising given the focus on improving tasks of the given domain for domain-specific models, rather than to improve in general tasks.

## 9 Conclusion

Despite the extensive prior work evaluating social bias in MLMs, the relationship between social biases and the factors associated with MLMs remains unclear. To address this gap, we conducted the first-ever comprehensive study of this type through predictive factor analysis to investigate the impact of different factors on social bias and task performance in pretrained MLMs, considering 39 models with 30 factors. We found that model size, tokenization and training objectives are the three most important factors across tasks. In terms of social biases, domain-specific models do not appear to be more or less biased than general domains, while multilingual models, which are trained on corpora covering different languages and cultures, appear to be less socially biased than the monolingual ones.

## Acknowledgements

Jose Camacho-Collados and Yi Zhou are supported by a UKRI future leaders fellowship. Danushka Bollegala holds concurrent appointments as a Professor at University of Liverpool and as an Amazon Scholar. This paper describes work performed at the University of Liverpool and is not associated with Amazon.

## Limitations

This paper studies the impact of underlying factors of MLMs on social bias and downstream task performance. In this section, we highlight some of the important limitations of this work. We hope this will be useful when extending our work in the future by addressing these limitations.

As described in § 6, the regression models we take into account in this paper are not able to be properly trained based on the features that we considered. Extending the factors and further exploring the reason for the poor performance of regression models on GLUE is one future direction.

We limit our work in this paper to focusing on evaluating intrinsic social bias captured by MLMs. However, there are numerous extrinsic bias evaluation datasets existing such as BiasBios (De-Arteaga et al., 2019), STS-bias (Webster et al., 2020), NLI-bias (Dev et al., 2020). Extending our work to evaluate the extrinsic biases in MLMs will be a natural line of future work.

Furthermore, our analysis focuses on MLMs and not considering generative language models such as GPT-2 (Radford et al.), Transformer-XL (Dai et al., 2019), XLNet (Yang et al., 2019) and GPT-3 (Brown et al., 2020). Extending our work to investigate the relationship between models' factors and social bias as well as downstream performance is deferred to future work.

Finally, although we tried to collect as many MLMs as possible, the final number may be in some cases insufficient to draw conclusive numerical conclusions in the regression analysis.

## Ethical Considerations

In this paper, we aim to investigate which factors affect the social bias captured by MLMs. Although we used existing datasets that are annotated for social biases, we did not annotate nor release new datasets as part of this research. In particular, we did not annotate any datasets by ourselves in this work and used multiple corpora and benchmark datasets that have been collected, annotated and repeatedly used for evaluations in prior works. To the best of our knowledge, no ethical issues have been reported concerning these datasets.

The gender biases considered in the bias evaluation datasets in this paper cover only binary gender (Dev et al., 2021). However, non-binary genders are severely underrepresented in textual data used to train MLMs. Moreover, non-binary genders are often associated with derogatory adjectives. Evaluating social bias by considering non-binary gender is important.

Furthermore, biases are not limited to word representations but also appear in sense representations (Zhou et al., 2022). However, our analysis did not include any sense embedding models.

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

# A Performance on All MLMs

## A.1 Performance of Regression Models over All MLMs

The comparison of different regression models using the features from all 39 MLMs is shown in Table 11. The $R^2$ scores obtained by all the regression models are not as high as the ones when only taking into account the monolingual MLMs in the general domain. On the other hand, when considering all the MLMs, the decision tree model obtains higher $R^2$ scores than gradient boosting.

| | Models | A-CP | A-SS | TweetEval | GLUE |
|---|---|---|---|---|---|
| $R^2$ | Gradient Boosting | 0.298 | 0.231 | 0.299 | -0.791 |
| | SVM | -0.741 | 0.066 | 0.185 | 0.022 |
| | Gaussian Process | -0.548 | -0.039 | **0.574** | -0.032 |
| | Decision Tree | **0.476** | **0.436** | 0.340 | -0.999 |
| | Random Forest | -0.328 | 0.314 | 0.401 | 0.027 |
| | Linear Regression | -1.793 | -1.761 | -0.247 | -5.184 |
| RMSE | Gradient Boosting | 2.963 | 3.071 | 2.726 | 5.862 |
| | SVM | 4.667 | 3.385 | 2.939 | 4.333 |
| | Gaussian Process | 4.401 | 3.570 | **2.125** | 4.449 |
| | Decision Tree | **2.560** | **2.631** | 2.645 | 6.194 |
| | Random Forest | 4.076 | 2.901 | 2.520 | **4.321** |
| | Linear Regression | 5.911 | 5.820 | 3.635 | 10.893 |

Table 11: Comparison of different regression models using the features from all the MLMs. The highest $R^2$ (on the top) and the lowest RMSE (on the bottom) are shown in bold.

## A.2 Important Scores of Factors for Training Gradient Boosting Models

Table 12 shows the full table of important scores for each of the factors used for training gradient boosting regression models over all 39 MLMs without removing the factors that obtain the important score of 0.

| Factors | A-CP | A-SS | TweetEval | GLUE |
|---|---|---|---|---|
| Parameter Size | **0.297** | 0.134 | 0.244 | 0.200 |
| Distill | 0.007 | 0.001 | 0.005 | 0.002 |
| Train-Books | 0.040 | 0.003 | 0.006 | 0.003 |
| train-Wiki | 0.000 | 0.000 | 0.000 | 0.000 |
| Train-CCNews | 0.005 | 0.001 | 0.014 | 0.001 |
| Train-OWT | 0.004 | 0.002 | 0.015 | 0.000 |
| Train-Stories | 0.004 | 0.001 | 0.026 | 0.003 |
| train-Tweets | 0.000 | 0.000 | 0.000 | 0.000 |
| train-RALE | 0.000 | 0.000 | 0.000 | 0.000 |
| train-Patent | 0.000 | 0.000 | 0.000 | 0.000 |
| train-Caselaw | 0.000 | 0.000 | 0.000 | 0.000 |
| train-GooglePatents | 0.000 | 0.000 | 0.000 | 0.000 |
| train-S2ORC | 0.000 | 0.000 | 0.000 | 0.000 |
| train-MIMIC3 | 0.000 | 0.000 | 0.000 | 0.000 |
| train-PMed | 0.000 | 0.000 | 0.000 | 0.000 |
| train-PMC | 0.000 | 0.000 | 0.000 | 0.000 |
| train-CC100 | 0.000 | 0.000 | 0.000 | 0.000 |
| Cont-train | 0.037 | 0.001 | 0.072 | 0.007 |
| Uncased | 0.071 | 0.080 | 0.007 | 0.035 |
| Domain | 0.000 | 0.000 | 0.000 | 0.000 |
| Language | 0.000 | 0.000 | 0.000 | 0.000 |
| # Languages | 0.000 | 0.000 | 0.000 | 0.000 |
| Vocabulary Size | 0.005 | 0.000 | 0.016 | 0.001 |
| Tokenization | 0.191 | **0.356** | 0.019 | 0.059 |
| # Layers | 0.056 | 0.047 | 0.093 | **0.520** |
| MLM | 0.136 | 0.115 | 0.024 | 0.008 |
| WWM | 0.162 | 0.169 | 0.015 | 0.013 |
| NSP | 0.001 | 0.041 | 0.125 | 0.047 |
| SOP | 0.002 | 0.052 | **0.256** | 0.100 |
| MRP | 0.023 | 0.001 | 0.071 | 0.005 |

Table 12: The important scores for each of the factors for training the *gradient boosting regression* model over all the MLMs. The highest importance scores are shown in bold, and the second highest ones are underlined. Due to the page limitation, we do not show the factors that obtain 0 important scores for all the tasks. MLM, WWM, NSP, SOP, and MRP represent masked language modelling, whole word masking, next sentence prediction, sentence ordering prediction, and mention reference prediction, respectively.

## A.3 Important Scores of Factors for Training Decision Tree Models

Table 13 shows the full table of important scores for each of the factors used for training decision tree models over all 39 MLMs. We derive similar conclusions regarding the most predictive factors. Specifically, parameter size is the most predictive factor on A-CP and the second most predictive factor on TweetEval. Tokenization and number of layers are the most predictive factors on A-SS and GLUE, respectively.

| Factors | A-CP | A-SS | TweetEval | GLUE |
|---|---|---|---|---|
| Parameter Size | **0.308** | 0.110 | 0.230 | 0.137 |
| Distill | 0.000 | 0.000 | 0.000 | 0.004 |
| Train-Books | 0.000 | 0.000 | 0.000 | 0.000 |
| train-Wiki | 0.000 | 0.000 | 0.000 | 0.000 |
| Train-CCNews | 0.000 | 0.000 | 0.172 | 0.000 |
| Train-OWT | 0.054 | 0.000 | 0.000 | 0.000 |
| Train-Stories | 0.000 | 0.000 | 0.000 | 0.000 |
| train-Tweets | 0.000 | 0.000 | 0.000 | 0.000 |
| train-RALE | 0.000 | 0.000 | 0.000 | 0.000 |
| train-Patent | 0.000 | 0.000 | 0.000 | 0.000 |
| train-Caselaw | 0.000 | 0.000 | 0.000 | 0.000 |
| train-GooglePatents | 0.000 | 0.000 | 0.000 | 0.000 |
| train-S2ORC | 0.000 | 0.000 | 0.000 | 0.000 |
| train-MIMIC3 | 0.000 | 0.000 | 0.000 | 0.000 |
| train-PMed | 0.000 | 0.000 | 0.000 | 0.000 |
| train-PMC | 0.000 | 0.000 | 0.000 | 0.000 |
| train-CC100 | 0.000 | 0.000 | 0.000 | 0.000 |
| Cont-train | 0.000 | 0.000 | 0.000 | 0.000 |
| Uncased | 0.096 | 0.085 | 0.013 | 0.045 |
| Domain | 0.000 | 0.000 | 0.000 | 0.000 |
| Language | 0.000 | 0.000 | 0.000 | 0.000 |
| # Languages | 0.000 | 0.000 | 0.000 | 0.000 |
| Vocabulary Size | 0.000 | 0.000 | 0.000 | 0.000 |
| Tokenization | 0.211 | **0.390** | 0.000 | 0.004 |
| # Layers | 0.004 | 0.083 | 0.160 | **0.518** |
| MLM | 0.000 | 0.000 | 0.000 | 0.000 |
| WWM | 0.289 | 0.278 | 0.015 | 0.072 |
| NNSP | 0.000 | 0.000 | 0.000 | 0.000 |
| SOP | 0.000 | 0.000 | **0.308** | 0.172 |
| MRP | 0.038 | 0.054 | 0.103 | 0.047 |

Table 13: The important scores for each of the factors for training the *decision tree* model over all the MLMs. The highest importance scores are shown in bold, and the second highest ones are underlined. Due to the page limitation, we do not show the factors that obtain 0 important scores for all the tasks. MLM, WWM, NSP, SOP, and MRP represent masked language modelling, whole word masking, next sentence prediction, sentence ordering prediction, and mention reference prediction, respectively.

## A.4 Comparison of All MLMs in Different Categories

Table 14 shows the comparison of social bias and task performance of MLMs on different tasks, according to different features associated with the important factors. For model size, we observe that the MLMs trained with a parameter size greater than 100M tend to have less social bias compared with the ones trained with a smaller parameter size while reporting competitive performance in both TweetEval and GLUE. The models trained with 6 and 12 layers contain less social bias with generally high downstream task performance. As for vocabulary size, the models trained with a medium-size vocabulary (i.e., greater than 50K and less

| | A-CP | A-SS | TweetEval | GLUE |
|---|---|---|---|---|
| **Parameter Size** | | | | |
| S ($x < 100M$) | 54.365 | 56.718 | 59.903 | **78.827** |
| M ($100M \leq x \leq 300M$) | **53.329** | 55.558 | 60.547 | 76.760 |
| L ($x > 300M$) | 53.598 | **55.437** | **61.895** | 78.377 |
| **Number of Layers** | | | | |
| 6 layers | 53.383 | **55.097** | 60.116 | **76.288** |
| 12 layers | **53.368** | 55.560 | **60.280** | 76.019 |
| 24 layers | 53.975 | 56.085 | 59.994 | 75.564 |
| **Vocabulary Size** | | | | |
| S ($x < 50K$) | 53.921 | 55.488 | 59.55 | 77.81 |
| M ($50K \leq x \leq 100K$) | 54.388 | 57.801 | **63.36** | **78.18** |
| L ($x > 100K$) | **51.107** | **52.370** | 59.71 | 74.83 |
| **Tokenization** | | | | |
| BPE | **53.457** | 56.419 | **62.247** | **77.674** |
| WordPiece | 53.559 | **55.053** | 59.685 | 77.392 |
| SentencePiece | 53.912 | 55.612 | 60.470 | 77.210 |
| **Training Objectives** | | | | |
| MLM | 53.714 | 56.673 | **63.174** | 77.115 |
| MLM + NSP | 52.599 | **53.579** | 59.171 | 76.498 |
| MLM + SOP | 56.995 | 59.185 | 57.314 | **84.926** |
| MLM + MRP | **52.140** | 56.230 | 60.012 | 79.872 |
| WWM + NSP | 59.715 | 60.805 | 62.225 | 80.420 |

Table 14: Comparison of social bias and task performance of MLMs on different tasks, according to different features associated with the important factors. MLM, NSP, WWM, SOP and MRP represent masked language modeling, whole word masking, next sentence prediction, sentence ordering prediction and mention reference prediction, respectively

than 100K) obtain the best performance on both TweetEval and GLUE, while the ones with a large vocabulary size (i.e., greater than 100K) contain lesser degrees of social biases. Regarding the to-kenization methods, we see that the models using BPE contain the lowest level of social bias (i.e., A-CP close to 50) and the best performance on both TweetEval and GLUE, while WordPiece is better for A-SS. Models using SentencePiece to-kenization perform consistently worse across the board, but we should also note the limited number of language models analysed with this tokenization method.

## B  Error Analysis on GLUE

From Table 15, we observe that CoLA, MRPC and RTE obtain negative $R^2$ scores. This indicates gradient boosting is not able to be properly trained to predict the performance of models on CoLA, MRPC and RTE given the considered features. However, the GLUE score is computed by taking the average of the performance over the 9 subtasks. The non-predictive subtasks especially on RTE task result in a negative $R^2$ value when training gradient boosting to predict the perfor-

| CoLA | SST-2 | MRPC | STS-B | QQP | MNLI | QNLI | RTE | WNLI |
|---|---|---|---|---|---|---|---|---|
| -0.070 | 0.294 | -1.339 | 0.122 | 0.125 | 0.648 | 0.032 | -11.793 | 0.182 |

Table 15: $R^2$ scores returned by gradient boosting on each GLUE sutasks.

mance of the MLM on GLUE.

## C Details of Factors

Table 16 shows the description of each factor including the feature types as well as the potential value for each factor. As shown in the table, most of the factors are binary, while factors such as parameter size, domain, language, number of languages, vocabulary size, tokenization and number of layers are categorical. Table 17 provides the details of factors corresponding to each language model.

| Factors | Description |
|---|---|
| Parameter Size | categorical, the parameter size is divided into three categories: S ($x < 100M$), M ($100M \leq x \leq 300M$), L ($x > 300M$) |
| Distill | binary, 1 if the model uses model distillation, otherwise 0 |
| Train-Books | binary, 1 if BookCorpus is used as a training corpus, otherwise 0 |
| Train-Wiki | binary, 1 if Wikipedia is used as a training corpus, otherwise 0 |
| Train-CCNews | binary, 1 if CC-News is used as a training corpus, otherwise 0 |
| Train-OWT | binary, 1 if OpenWebText is used as a training corpus, otherwise 0 |
| Train-Stories | binary, 1 if Stories is used as a training corpus, otherwise 0 |
| Train-Tweets | binary, 1 if Tweets is used as a training corpus, otherwise 0 |
| Train-RALE | binary, 1 if RALE is used as a training corpus, otherwise 0 |
| Train-Patent | binary, 1 if Patent Litigations is used as a training corpus, otherwise 0 |
| Train-Caselaw | binary, 1 if Caselaw Access Project is used as a training corpus, otherwise 0 |
| Train-GooglepPatents | binary, 1 if Google Patents Public Data is used as a training corpus, otherwise 0 |
| Train-S2ORC | binary, 1 if S2ORC is used as a training corpus, otherwise 0 |
| Train-MIMIC3 | binary, 1 if MIMIC3 is used as a training corpus, otherwise 0 |
| Train-PMed | binary, 1 if PubMed Abstracts is used as a training corpus, otherwise 0 |
| Train-PMC | binary, 1 if PMC Full-text articles are used as a training corpus, otherwise 0 |
| Train-CC100 | binary, 1 if CommonCrawl data in 100 language (CC100) is used as a training corpus, otherwise 0 |
| Cont-train | binary, 1 if the model is continued trained based on a pretrained MLMs, otherwise 0 |
| Uncased | binary, 1 if the model is uncased, otherwise 0 |
| Domain | categorical, domain of the models is based on the domain of training corpus that the model used during pretraining. The 4 domains are considered: general domain, social media, legal domain, biomedical domain |
| Language | categorical, English only or multilingual |
| Number of Languages | categorical, we take the real number of the languages that are trained on |
| Vocabulary size | categorical, the vocabulary size is divided into three categories: S ($x < 50K$), M ($50K \leq x \leq 100K$), L ($x > 100K$) |
| Tokenization | categorical, three categories: BPE, WordPiece, SentencePiece |
| Number of Layers | categorical, three categories: 6 layers, 12 layers, 24 layers |
| Masked Language Modelling | binary, 1 if the model uses the masking technique, otherwise 0 |
| Whole Word Masking | binary, 1 if the model uses the whole word masking technique, otherwise 0 |
| Next Sentence Prediction | binary, 1 if the model uses next sentence prediction objective, otherwise 0 |
| Sentence Order Prediction | binary, 1 if the model uses sentence order prediction objective, otherwise 0 |
| Mention Reference Prediction | binary, 1 if the model uses mention reference prediction objective, otherwise 0 |

Table 16: The description of the features of each factor that we consider in this paper.

| Models | Para | Distill | Training Corpora | Cont | UnC | Domain | Lang | # Lang | Vob | Tokenization | # Layers | Objectives |
|---|---|---|---|---|---|---|---|---|---|---|---|---|
| roberta-base | 125M | 0 | BookCorpus, English Wikipedia, CCNews OpenWebText, Stories | 0 | 0 | General | English | 1 | 50K | BPE | 12 | MLM |
| roberta-large | 355M | 0 | BookCorpus, English Wikipedia, CCNews OpenWebText, Stories | 0 | 0 | General | English | 1 | 50K | BPE | 24 | MLM |
| bert-base-cased | 110M | 0 | BookCorpus, English Wikipedia | 0 | 0 | General | English | 1 | 30K | WordPiece | 12 | MLM, NSP |
| bert-large-cased | 340M | 0 | BookCorpus, English Wikipedia | 0 | 0 | General | English | 1 | 30K | WordPiece | 24 | MLM, NSP |
| bert-base-uncased | 110M | 0 | BookCorpus, English Wikipedia | 0 | 1 | General | English | 1 | 30K | WordPiece | 12 | MLM, NSP |
| bert-large-uncased | 340M | 0 | BookCorpus, English Wikipedia | 0 | 1 | General | English | 1 | 30K | WordPiece | 24 | MLM, NSP |
| bert-large-cased-whole-word-masking | 336M | 0 | BookCorpus, English Wikipedia | 0 | 0 | General | English | 1 | 30K | WordPiece | 24 | WWM, NSP |
| bert-large-uncased-whole-word-masking | 336M | 0 | BookCorpus, English Wikipedia | 0 | 1 | General | English | 1 | 30K | WordPiece | 24 | WWM, NSP |
| albert-base-v2 | 11M | 0 | BookCorpus, English Wikipedia | 0 | 0 | General | English | 1 | 30K | WordPiece | 12 | MLM, NSP |
| albert-large-v2 | 17M | 0 | BookCorpus, English Wikipedia | 0 | 0 | General | English | 1 | 30K | WordPiece | 24 | MLM, NSP |
| distilbert-base-cased | 66M | 0 | BookCorpus, English Wikipedia | 0 | 0 | General | English | 1 | 30K | WordPiece | 6 | MLM, NSP |
| distilbert-base-uncased | 66M | 1 | BookCorpus, English Wikipedia | 0 | 1 | General | English | 1 | 30K | WordPiece | 6 | MLM, NSP |
| distilroberta-base | 82M | 1 | BookCorpus, English Wikipedia, CCNews OpenWebText, Stories | 0 | 0 | General | English | 1 | 50K | BPE | 6 | Masked |
| albert-xxlarge-v2 | 223M | 0 | BookCorpus, English Wikipedia | 0 | 0 | General | English | 1 | 30K | SentencePiece | 12 | MLM, SOP |
| albert-xlarge-v2 | 58M | 0 | BookCorpus, English Wikipedia | 0 | 0 | General | English | 1 | 30K | SentencePiece | 24 | MLM, SOP |
| nielsr/coref-roberta-large | 355M | 0 | BookCorpus, English Wikipedia, CCNews OpenWebText, Stories | 1 | 0 | General | English | 1 | 50K | BPE | 24 | MLM, MRP |
| nielsr/coref-roberta-base | 125M | 0 | BookCorpus, English Wikipedia, CCNews OpenWebText, Stories | 1 | 0 | General | English | 1 | 50K | BPE | 12 | MLM, MRP |
| nielsr/coref-bert-base | 110M | 0 | BookCorpus, English Wikipedia | 1 | 0 | General | English | 1 | 30K | WordPiece | 12 | MLM, MRP |
| nielsr/coref-bert-large | 340M | 0 | BookCorpus, English Wikipedia | 1 | 0 | General | English | 1 | 30K | WordPiece | 24 | MLM, MRP |
| cardiffnlp/twitter-roberta-base | 125M | 0 | BookCorpus, English Wikipedia, CCNews OpenWebText, Stories, English tweets | 1 | 0 | Social Media | English | 1 | 50K | BPE | 12 | MLM |
| cardiffnlp/twitter-scratch-roberta-base | 125M | 0 | BookCorpus, English Wikipedia, CCNews OpenWebText, Stories, 58M tweets | 0 | 0 | Social Media | English | 1 | 50K | BPE | 12 | MLM |
| vinai/bertweet-base | 110M | 0 | 845M English Tweets streamed from 01/2012 to 08/2019 and 5M Tweets related to the COVID-19 pandemic | 0 | 0 | Social Media | English | 1 | 64K | BPE | 12 | MLM |
| vinai/bertweet-large | 340M | 0 | 845M English Tweets streamed from 01/2012 to 08/2019 and 5M Tweets related to the COVID-19 pandemic | 0 | 0 | Social Media | English | 1 | 64K | BPE | 24 | MLM |
| GroNLP/hateBERT | 110M | 0 | BookCorpus, English Wikipedia, RALE (Reddit) | 1 | 0 | Social Media | English | 1 | 34K | WordPiece | 12 | MLM, NSP |
| cardiffnlp/twitter-roberta-large-2022-154m | 355M | 0 | BookCorpus, English Wikipedia, CCNews OpenWebText, Stories, English tweets | 1 | 0 | Social Media | English | 1 | 50K | BPE | 24 | MLM |
| cardiffnlp/twitter-roberta-base-2022-154m | 125M | 0 | BookCorpus, English Wikipedia, CCNews OpenWebText, Stories, English tweets | 1 | 0 | Social Media | English | 1 | 50K | BPE | 12 | MLM |
| saibo/legal-roberta-base | 125M | 0 | BookCorpus, English Wikipedia, CCNews OpenWebText, Stories, Patent Litigations, Caselaw Access Project, Google Patents Public Data | 1 | 0 | Legal | English | 1 | 50K | BPE | 12 | MLM |
| allenai/biomed_roberta_base | 125M | 0 | BookCorpus, English Wikipedia, CCNews OpenWebText, Stories, 2.68M full-text biomedical papers from S2ORC | 1 | 0 | Biomedical | English | 1 | 50K | BPE | 12 | MLM |
| emilyalsentzer/Bio_ClinicalBERT | 110M | 0 | BookCorpus, English Wikipedia, MIMIC3 | 1 | 0 | Biomedical | English | 1 | 30K | WordPiece | 12 | MLM, NSP |
| dmis-lab/biobert-base-cased-v1.2 | 110M | 0 | BookCorpus, English Wikipedia, PubMed Abstracts and PMC Full-text articles | 1 | 0 | Biomedical | English | 1 | 30K | WordPiece | 12 | MLM, NSP |
| bionlp/bluebert_pubmed_mimic_uncased_L-12_H-768_A-12 | 110M | 0 | BookCorpus, English Wikipedia, MIMIC3 and PubMed abstracts | 1 | 1 | Biomedical | English | 1 | 31K | WordPiece | 12 | MLM, NSP |
| bionlp/bluebert_pubmed_mimic_uncased_L-24_H-1024_A-16 | 340M | 0 | BookCorpus, English Wikipedia, MIMIC3 and PubMed abstracts | 1 | 1 | Biomedical | English | 1 | 31K | WordPiece | 24 | MLM, NSP |
| bert-base-multilingual-cased | 177M | 0 | Wikipedias | 0 | 0 | General | Multilingual | 104 | 120K | BPE | 12 | MLM, NSP |
| bert-base-multilingual-uncased | 177M | 0 | Wikipedias | 0 | 1 | General | Multilingual | 104 | 120K | BPE | 12 | MLM, NSP |
| distilbert-base-multilingual-cased | 134M | 1 | Wikipedias | 0 | 0 | General | Multilingual | 104 | 120K | BPE | 12 | MLM, NSP |
| xlm-roberta-base | 270M | 0 | 2.5TB of filtered Common Crawl data (CC100) | 0 | 0 | General | Multilingual | 100 | 250K | SentencePiece | 12 | MLM |
| xlm-roberta-large | 550M | 0 | 2.5TB of filtered Common Crawl data (CC100) | 0 | 0 | General | Multilingual | 100 | 250K | SentencePiece | 24 | MLM |
| facebook/xlm-v-base | 891M | 0 | 2.5TB of filtered Common Crawl data (CC100) | 0 | 0 | General | Multilingual | 100 | 1M | SentencePiece | 12 | MLM |
| cardiffnlp/twitter-xlm-roberta-base | 270M | 0 | 198M multilingual tweets | 1 | 0 | Social Media | Multilingual | 100 | 1.724M | SentencePiece | 12 | MLM |

Table 17: The details of factors corresponding to each model. Cont and UnC represent continual training and uncased, respectively. MLM, NSP, WWM, SOP, and MRP represent masked language modelling, next sentence prediction, whole word masking, sentence ordering prediction, and mention reference prediction, respectively.