# OpenReview forum: "A Predictive Factor Analysis of Social Biases and Task-Performance in Pretrained Masked Language Models"
_EMNLP/2023/Conference — EMNLP 2023 Main_

### Official Review · Reviewer_Swdc · 2023-07-31

**Typos Grammar Style And Presentation Improvements:** Textural -> textual (Introduction)
**Soundness:** 4

**Excitement:**

4: Strong: This paper deepens the understanding of some phenomenon or lowers the barriers to an existing research direction.

**Paper Topic And Main Contributions:**

This work aims to measure the impact of model attributes (size, training data, tokenization methodology, pre-training objective) on the prevalence of social biases in MLMs. To do so, the authors propose the use of predictive factor analysis, considering 39 models with 30 factors. The models used for analysis are monolingual/general, monolingual/domain-specific, and multi-lingual/general MLMs. The factors include model size (varying from less than 100M params to greater than 300M params), training objectives (whole word masking, subword masking, sentence ordering prediction, mention reference prediction), training corpora (general domain, social media, legal domain, academia domain, biomedical domain, multilingual), tokenization techniques (BPE, WordPiece, SentencePiece), and language (monolingual, multi-lingual). The bias scores for each model were computed using two prevalent bias benchmark (StereoSet and Crow-S-Pairs). General downstream performance was computed using GLUE and TweetEval. In order to conduct the analysis, the factors were fed into a regression model as input for training, using both bias score and task performance as output. The results indicate that model size, tokenization and training objectives are the three most consequential factors and multilingual models appeared to be less socially-biased than the monolingual ones.

**Questions For The Authors:**

(1) Why was the research design observational rather than experimental, and how does it limit the ability to establish causal relationships between model attributes and social biases in MLMs?

(2) Can the consideration of multiple factors simultaneously (e.g., model size, training objectives, corpora, tokenization techniques, and language) lead to complex interactions and potential confounding effects?

**Reasons To Accept:**

The main strengths of this paper pertains to the diversity of model types (monolingual/general, monolingual/domain-specific, and multi-lingual/general MLMs) and exhaustiveness of model factors (size, training data, tokenization methodology, pre-training objective) within the experimental setup. The authors also utilize two widely known bias benchmarks (StereoSet and Crow-S-Pair) while adapting their standard evaluation protocols to use a more representative metric (AULA), leveraging attention weights to encode token importance (robust against frequency biases).

**Reasons To Reject:**

The main weaknesses of this paper pertains to the research design (observational rather than experimental), which limits the ability to establish causal relationships between model attributes and the prevalence of social biases in MLMs. Without experimental manipulation of variables, it is challenging to determine whether the identified correlations are genuinely indicative of cause-and-effect relationships. Additionally, the study considers multiple factors simultaneously, such as model size, training objectives, corpora, tokenization techniques, and language, which can lead to complex interactions and potential confounding effects. The inability of controlling all potential confounding variables is understandable due to training cost constraints, however, the authors should be cautious when interpreting the results and consider these limitations when drawing conclusions (challenging to establish causation). Finally, the authors excluded de-biased models (CDA, Dropout, Context-Debias, Auto-Debias) as well as a prevalent bias benchmark (SEAT) within the experimental setup.

**Reproducibility:**

4: Could mostly reproduce the results, but there may be some variation because of sample variance or minor variations in their interpretation of the protocol or method.

**Reviewer Confidence:**

5: Positive that my evaluation is correct. I read the paper very carefully and I am very familiar with related work.

---

> ### Author Rebuttal · Authors · 2023-08-28
>
> **RR1/Q1:** Why was the research design observational rather than experimental, and how does it limit the ability to establish causal relationships between model attributes and social biases in MLMs?
>
> **Response:** As correctly pointed out an experimental setting where we experimentally change a single independent variable, keeping all other variables fixed (i.e. treating them as controlled variables) would be the ideal scenario for making straightforward conclusions. Unfortunately, however, retraining large MLMs is expensive, which forces us to resort to the observational analysis of existing pretrained MLMs, as conducted in this paper. However, one of the main methodological contributions of this paper is to demonstrate how such an observational analysis could be successfully conducted using existing pretrained MLMs. Given that many MLMs are trained and publicly released by different researchers in NLP/ML, we consider the ability to perform such an observational analysis in a cost-effective manner will be of interest to the wider EMNLP audience.
>
> **RR2/Q2:** Can the consideration of multiple factors simultaneously lead to complex interactions and potential confounding effects?
>
> **Response:** Indeed as we explicitly mentioned under challenge (a) in the introduction (Lines 46-50), we recognise this problem and propose to use gradient boosting regression, which considers the non-linear interactions among the underlying factors when predicting the given target variable (e.g. social bias scores, GLUE scores, etc.). Moreover, in order to further limit this potential issue on similar features, we cluster the features into high-level categories.
>
> **RR3:** The authors excluded de-biased models (CDA, Dropout, Context-Debias, Auto-Debias) as well as a prevalent bias benchmark (SEAT) within the experimental setup.
>
> **Response:** We agree that analysing the effect of debiasing methods is an interesting research direction. However, our goal in the paper is to investigate the underlying factors of pretrained MLMs. Therefore, we did not consider any debiased models as not to interfere with the main analysis. For evaluation, we decided to use two recent and commonly used benchmark datasets, CrowS-Pairs [Nangia+ 2020] and StereoSet [Nadeem+ 2021], because, unlike SEAT which is extended on word association datasets, these are explicitly designed to measure social bias on pretrained masked language models.

---

### Official Review · Reviewer_QwCm · 2023-08-04

**Soundness:** 3

**Excitement:**

4: Strong: This paper deepens the understanding of some phenomenon or lowers the barriers to an existing research direction.

**Paper Topic And Main Contributions:**

The paper studies 39 pretrained Masked Language Models (MLMs) by identifying 30 factors and modeling the correlation between these factors and several benchmarks ranging from social bias, GLUE, and TweetEval. After determining the best predictive model from those ~30 factors to score, they identify which factors (features) are most important. They conclude that model parameters size and tokenization are typically the most important factors, followed by training objectives. In subsequent analysis, they compare all models of each class to, e.g., demonstrate that larger models score better on the benchmarks.

**Questions For The Authors:**

A. For this regression analysis, is there theory/significance testing around this? Is the R^2 value good even though it is negative for GLUE? Even if this is a reliable model/significant R^2, some discussion confirming this would help the reader in contextualizing the results and their confidence in the strength of the findings.

B. Do the other models with lower R^2 (e.g. SVM, decision tree, RFs) also point to the same conclusions regarding the most predictive factors? If so, that lends a little more credibility to the conclusions of the work.

C. In Table 9, why does the small model do so well on GLUE? Is this due to small sample sizes or is there something anomalous (or absent) in the “S” split which is causing this aberrant behavior?

D. Are the bold numbers in Table 9/10 significant? Which ones, if any, are?

E. How does this analysis be related to SHAP value analysis (and why wasn’t SHAP value analysis considered/discussed?)

**Reasons To Accept:**

* The posed question of determining the factors that affect social biases and subsequent methodology is an interesting one to explore, well-motivated as a natural “scaled up” version of similar prior work.
* The conclusions drawn from the paper are insightful directions for future work and model development in the pretrained MLM space.

**Reasons To Reject:**

* The regression analysis, which is one of the core components of the paper, is difficult to interpret (e.g. R^2 for GLUE is always negative), leading to unclearness around whether the results/analysis of the factors are significant or meaningful; this is also mentioned in the Limitations.
  *  As one example, I would imagine distillation should have some consistently correlated impact on GLUE, but its importance is only 0.002?
* While the appendix includes a description of all the factors and what each value means, it should also include all the models, how the authors categorized them, and the number of models with each value, for each factor. In particular, there is no mention of significance in any of the results, and maybe some categories are much smaller than others, which may lead to skewed results.

**Reproducibility:**

2: Would be hard pressed to reproduce the results. The contribution depends on data that are simply not available outside the author's institution or consortium; not enough details are provided.

**Reviewer Confidence:**

3: Pretty sure, but there's a chance I missed something. Although I have a good feel for this area in general, I did not carefully check the paper's details, e.g., the math, experimental design, or novelty.

**Typos Grammar Style And Presentation Improvements:**

L468: “nd” -> “and”

L588-590 is unclear: models with 6 layers show a comparable capability to what? Reduced social bias or better performance on downstream tasks?

When there is more space, or in the appendix, please include the sizes of the datasets used.

As mentioned, please include how what values each model's factors were labeled in the appendix, and ideally (commit to) releasing code too.

---

> ### Author Rebuttal · Authors · 2023-08-28
>
> **RR1:** The regression analysis is difficult to interpret (e.g. R^2 for GLUE is always negative) -> As one example, I would imagine distillation should have some consistently correlated impact on GLUE.
>
> **Response:** As already stated above in the response to Reviewer 1, we use a regression model to analyse the impact of factors because those factors are not independent. By using a non-linear regression model (such as the gradient boosted tree regression we use in the paper) we are able to take into account the dependencies among the underlying factors.
>
> As mentioned in Lines 470-472 (and also in the Limitations section) R^2 for GLUE is always negative, which indicates that a regression model cannot be properly trained based on the features that we consider in this paper for predicting GLUE scores. Moreover, the important scores in Table 5 show the distribution of the important factors (i.e. the sum of the important score for each column is 1). We will clarify this to better explain the results.
>
> Distillation indeed has non-zero important scores, indicating that distillation is an important factor for predicting GLUE score. However, distillation is not as predictive as other factors such as parameter size, tokeniser etc.
>
> **RR2:** A description of how the authors categorise models and the number of models for each factor.
>
> **Response:** Thank you for the suggestion. We will add this information in the Appendix as a table, showing which models belong to which category.
>
> **Q1:** Is the R^2 value good even though it is negative for GLUE? Even if this is a reliable model/significant R^2, some discussion confirming this would help the reader in contextualizing the results and their confidence in the strength of the findings.
>
> **Response:** Thank you for the helpful suggestion. Indeed, negative R^2 scores for GLUE indicate that a regression model cannot be properly trained based on the features that we consider for predicting GLUE scores. This is not the case for the social bias benchmarks, for example, in which the R^2 score is substantial in most cases and the importance of relative features can be properly assessed. We will make this clear in the final version and contextualize based on the R^2 values.
>
> **Q2:** Do the other models with lower R^2 also point to the same conclusions regarding the most predictive factors?
>
> **Response:**  The conclusions are largely similar. In the case of Random Forest, the most predictive factor on A-CP and TweetEval is the parameter size, while the most predictive factors on A-SS and GLUE are tokenization and the number of layers, respectively. These results agree with what is shown in Table 5, except that the parameter size is the second best on TweetEval in Table 5.
>
> Similarly, in the case of decision trees, parameter size is the most predictive factor on A-CP, and the second most predictive factor on TweetEval. Tokenization and number of layers are the most predictive factors on A-SS and GLUE, respectively.
>
> **Q3:** In Table 9, why does the small model do so well on GLUE?
>
> **Response:** Please note that the number of models in each category is different and this is the reason behind this result. Specifically, there are 21 MLMs in the Medium category and 12 in the Large category, whereas there are only 6 in the Small category. Because some outlier cases are affecting the overall averages in each class, we compute the top-5 performing models in each class. For example, for S (Small) the top-5 GLUE scores are 85.348, 79.924, 78.487, 75.699, 74.676, while those in L (Large) are 83.221, 82.862, 81.546, 80.323, 79.293. The averages of these top-performing models show that L (avg 81.449) models have better GLUE scores than S (avg 78.827). We will include this information in the final version with the extra page allowed.
>
> **Q4:** Are the bold numbers in Table 9/10 significant? Which ones, if any, are?
>
> **Response:** We conducted Clopper-Pearson exact tests to check whether the best performances (indicated by bold values)  for A-CP and A-SS are statistically significant. According to the Clopper-Pearson exact test, A-SS scores obtained by large models in the vocabulary size category are statistically significant in Table 9. Moreover, A-SS scores obtained by the models using MLM + NSP training objectives are also statistically significant. Furthermore, in Table 10, all the bold values for A-SS are also statistically significant.
>
> **Q5:** Why not use SHAP?
>
> **Response:** Please note that we have 30 factors in our analysis, which results in 2^29 - 1 = 536,870,911 coalitions (subsets). Although there are efficient approximations for SHAP calculations, we did not explore this path due to computational issues, and instead used R^2 and Gini importance in this work.

---

### Official Review · Reviewer_8eDx · 2023-08-09

**Soundness:** 3

**Excitement:**

3: Ambivalent: It has merits (e.g., it reports state-of-the-art results, the idea is nice), but there are key weaknesses (e.g., it describes incremental work), and it can significantly benefit from another round of revision. However, I won't object to accepting it if my co-reviewers champion it.

**Missing References:**

Related work w.r.t. methodology: The framework bears some similarity with work on prediction-based neural architecture search (NAS). In particular, [1] uses gradient boosting based on binary features to predict task accuracy (section 3). The main difference is that [1] solely uses architecture-related features and uses prediction as an intermediate step within search iterations; this work uses mostly pretraining-related features along with some architecture-related features and performs extensive analysis.

[1] https://arxiv.org/pdf/2007.04785.pdf

**Paper Topic And Main Contributions:**

The goal is to investigate the relationship between pretraining factors of masked language models and social bias & downstream performance.
- To this end, they first take 39 off-the-shelf models and evaluate them on Crowdsourced Stereotype Pairs (CP), SteroSet (SS), TweetEval, and GLUE benchmark.
- Then they perform regression analysis: 39 data points (i.e., models), 4 aforementioned metrics as response variables, and 30 categorical features are pretraining factors that cover five groups (model size, training methods, training corpora, tokenization, language).
  - They compare various regression models: Gradient Boosting, SVM, Gaussian Process Regression, Decision Tree, Random Forest, and Logistic Regression. The performance is measured by the coefficient of determination (R^2) and RMSE. They find that GLUE cannot be reliably predicted by all regression models, and that gradient boosting is the best overall fit for the other three metrics.
  - With gradient boosting, they compare the feature importance. The analysis highlights the importance of model size, tokenization, and training objectives.
  (1) element-wise analysis: gini importance score for each feature, RMSE drop by removing a single feature.
  (2) group-wise analysis: RMSE drop by removing a single group (out of five) of features, RMSE score by only retaining a single group.
- They also present an empirical comparison within some of the important features by direct averaging: while tokenization is important, they find that BPE is generally better than WordPiece or SentencePiece; multilingual models have lower social bias while monolingual models have better performance on English-only downstream tasks.

**Reasons To Accept:**

- the paper is well-written and easy to follow

- comprehensive regression analysis that covers a wide range of existing pretrained MLMs

**Reasons To Reject:**

- Possible overfit of nonlinear regression model: they use 39 data points (i.e., models) to fit 30 categorical features with gradient boosting. It does not evaluate the generalization ability (e.g., cross-validation).

- It is relatively hard to interpret the result from regression/averaging. Prior work usually uses side-by-side comparison, accompanied by statistical tests (e.g., bert vs. distill-bert, roberta vs. distill-roberta, etc), when analyzing categorical factors.

**Reproducibility:**

3: Could reproduce the results with some difficulty. The settings of parameters are underspecified or subjectively determined; the training/evaluation data are not widely available.

**Reviewer Confidence:**

2: Willing to defend my evaluation, but it is fairly likely that I missed some details, didn't understand some central points, or can't be sure about the novelty of the work.

**Typos Grammar Style And Presentation Improvements:**

- Line 468: nd RMSE -> and RMSE

- For comprehensiveness, it would be helpful if you can list 30 features (similar to Table 3) along with the performance for all 39 models in the appendix.

---

> ### Author Rebuttal · Authors · 2023-08-28
>
> **RR1:** Possible overfit of nonlinear regression model.
>
> **Response:** Response: Our goal in this work is to find factors that best explain the performance and social biases (if any) present in MLMs. In particular, we do not propose to predict such targets for unseen models (i.e. generalisation of the prediction model is not within the scope of this paper). On the other hand, if the factors we use can explain well (overfit to) the observed targets then that would be considered as more suitable.
>
> **RR2:** It is relatively hard to interpret the result from regression/averaging.  Prior work usually uses side-by-side comparison.
>
> **Response:** Please note that we use 39 models in our analysis, which makes it difficult to perform 39 x 38 / 2 = 741 side-by-side comparisons, which would also be difficult to interpret individually and when aggregated. For this reason, we group models by higher-level attributes (e.g. pre-train corpora, model size, tokeniser, etc.) and analyse trends that can be observed over those attributes via averaging.
>
> On the other hand, as we already mentioned in the introduction section of the paper, one of the challenges in our analysis is that the underlying factors are not independent. Therefore, we use non-linear regression methods such as gradient boosted tree regression to consider the dependencies among those different factors.
>
> **Missing Reference:** Related work on NAS
>
> **Response:** Thank you for pointing out this work on NAS. Although it is not focused on social biases in MLMs, which is the main focus of our work in this paper, we agree that it would be useful to cite this work for its relevance to the factor analysis we perform. Therefore, we will include this reference in the final version of the paper if accepted.

---

### Meta-Review · Area_Chair_CQJv · 2023-09-18

**Recommendation:** 2

**Metareview:**

This paper systematically studies the impact of a large number of diverse MLM properties on task performance and social biases, applying a factor analysis/regression framework. Its main contribution is the large-scale analysis (many factors, many model) enabled by the adopted analysis methodology.

Overall, this is an innovative paper, which suggests a promising methodology for the ever important task of understanding the impact of MLM design decisions on their performance and biases.

The main weakness is the poor fit of the regression model for the GLUE benchmark, which calls into question the validity of all following analyses on GLUE -- to the extent that I would suggest to remove the GLUE results from the paper. At the very leas, the paper needs to add clear warning messaging around the reliability of the GLUE results. I also suggest to check whether specific GLUE task(s) are responsible for the low R2 value.

I concur with all other concerns raised by the reviewers (and discussed subsequently) and take these into account.

---

### Decision · Program_Chairs · 2023-10-07

**Decision:**

Accept-Main

**Comment:**

This paper systematically studies the impact of a large number of diverse MLM properties on task performance and social biases, applying a factor analysis/regression framework. Its main contribution is the large-scale analysis (many factors, many model) enabled by the adopted analysis methodology.

Overall, this is an innovative paper, which suggests a promising methodology for the ever important task of understanding the impact of MLM design decisions on their performance and biases.

The main weakness is the poor fit of the regression model for the GLUE benchmark, which calls into question the validity of all following analyses on GLUE -- to the extent that I would suggest to remove the GLUE results from the paper. At the very leas, the paper needs to add clear warning messaging around the reliability of the GLUE results. I also suggest to check whether specific GLUE task(s) are responsible for the low R2 value.

I concur with all other concerns raised by the reviewers (and discussed subsequently) and take these into account.